# Safety and Tolerability of Concentrated Intraventricular Nicardipine for Poor-Grade Aneurysmal Subarachnoid Hemorrhage–Related Vasospasm

**DOI:** 10.3390/jpm13030428

**Published:** 2023-02-27

**Authors:** Kaneez Zahra, Ricardo A. Domingo, Marion T. Turnbull, Christan D. Santos, Sarah H. Peacock, Daniel A. Jackson, Rabih G. Tawk, Jason L. Siegel, William David Freeman

**Affiliations:** 1Department of Neurology, Mayo Clinic, Jacksonville, FL 32224, USA; 2Department of Neurologic Surgery, Mayo Clinic, Jacksonville, FL 32224, USA; 3Department of Critical Care Medicine, Mayo Clinic, Jacksonville, FL 32224, USA; 4Department of Pharmacy, Mayo Clinic, Jacksonville, FL 32224, USA; 5Department of Radiology, Mayo Clinic, Jacksonville, FL 32224, USA

**Keywords:** delayed cerebral ischemia, intraventricular nicardipine, subarachnoid hemorrhage

## Abstract

**Objective:** To report the preliminary safety, tolerability, and cerebral spinal fluid (CSF) sampling utility of serial injections of concentrated intraventricular nicardipine (IVN) in the treatment of aneurysmal subarachnoid hemorrhage (aSAH). **Methods:** We report the clinical, radiographic, and laboratory safety and tolerability data of a retrospective case series from a single academic medical center. All patients with aSAH developed vasospasm despite enteral nimodipine and received serial injections of concentrated IVN (2.5 mg/mL). CSF injection safety, tolerability, and utility are defined and reported. **Results:** A total of 59 doses of concentrated IVN were administered to three patients with poor-grade SAH. In Case 1, a 33-year-old man with modified Fisher scale (mFS) grade 4 and Hunt-Hess scale (HH) score 4 received 26 doses; in Case 2, a 36-year-old woman with mFS grade 4 and HH score 5 received 13 doses; and in Case 3, a 70-year-old woman with mFS grade 3 and HH score 4 received 20 doses. No major safety or tolerability events occurred. Two patients were discharged to a rehabilitation facility, and one died after discharge from the hospital. **Conclusions:** A concentrated 4 mg IVN dose (2.5 mg/mL) in a 1.6 mL injection appears relatively safe and tolerable and potentially offers a second-line strategy for treating refractory vasospasm in poor-grade SAH without compromising intracranial pressure or cerebral perfusion pressure.

## 1. Introduction

Despite decades of research, aneurysmal subarachnoid hemorrhage (aSAH) has only one drug, nimodipine, an L-type calcium channel antagonist, proven in multiple randomized trials to improve neurologic outcomes [1]. Delayed cerebral ischemia (DCI) [2] after aSAH occurs between days 4 and 14 after initial aneurysmal rupture. The US Food and Drug Administration package insert indicates that the mechanism for nimodipine is a neuroprotective drug based on randomized trials of clinical outcomes and that the drug does not work purely on vasospasm alone [3]. The time course for vasospasm overlaps greatly with DCI and may be mediated by spreading neuronal depolarization [4], which adds complexity whether aSAH secondary brain injury is primarily by vasospasm-mediated ischemia, delayed neuronal injury from spreading neuronal depolarization–mediated DCI, or a combination of both [5]. Secondary brain injury from DCI and vasospasm-attributed causes remains a major cause of morbidity in initial aSAH survivors [1,2]. Therefore, interventions that reverse secondary brain injury may provide potential clues to future neurotherapeutics, and a secondary complementary agent to nimodipine is desperately needed. The standard dose of nimodipine 60 mg every 4 h is reduced in nearly half of all patients with SAH [6]. Plasma concentrations of nimodipine also vary depending on the route of administration [7]. In the US, only enteral use of nimodipine is recommended, since a black box warning was issued after the pill contents of nimodipine were extracted and given intravenously, causing severe cardiovascular events and death [3,7,8]. Because of these and other adverse events, including hypotension from enteral nimodipine [9,10], intraventricular nicardipine (IVN) doses of 4 mg emerged as a second-line rescue drug in aSAH vasospasm management and with an existing external ventricular drain (EVD) [11,12,13].

We originally reported our experience with IVN injection in the treatment of aSAH using a large volume 1 mg:1 mL concentration [12] with the isovolemic technique and with about 6 mL (4 mg:4 mL of nicardipine and 2 mL of sterile flush) through an existing EVD. After that work, we realized a main limitation is that it is clinically challenging to aspirate 6 mL of cerebral spinal fluid (CSF) and inject that intracranial volume in some cases. Thus, we hypothesized that a super-concentrated form of the drug could be helpful to overcome these intracranial volume injection limitations, for those at risk for increased intracranial pressure (ICP), and to maintain cerebral perfusion pressure (CPP) [14,15,16]. The primary aim of this study was to assess the safety and tolerability of administration of concentrated IVN in patients with aSAH. 

## 2. Materials and Methods

### 2.1. Study Design 

We report a retrospective case series from a single academic medical center (Mayo Clinic in Jacksonville, Florida). Inclusion criteria includes patients with aSAH who received intraventricular nicardine for treatment of symptomatic vasospasm. We previously reported a 1 mg:1 mL IVN dosing for patients with aSAH [12], which required aspiration of 6 mL CSF from the EVD to inject 4 mg of IVN (4 mL) followed by 2 mL sterile preservative-free saline. All patients with aSAH who received IVN had an EVD placed per standard of care for symptomatic hydrocephalus (Figure 1A,B). 

### 2.2. IVN Concentration and Injection Methodology

We measured the volume required to flush the EVD system from the proximal 3-way port to the intracranial EVD tip being equal to 2 mL using a Codman Bactiseal 1.9 mm inner diameter ventricular catheter. All patients had a Becker external drainage and monitoring system (Medtronic) with zero leveled at the tragus (Figure 1C). The sterile isovolemic injection apparatus is shown in Figure 1D, after making a sterile field around the EVD injection port. The sterile injection apparatus needle is inserted into the patient’s EVD proximal rubber port and CSF aspirated using a 3-way stopcock, followed by IVN drug injected intracranially (Figure 1D). This is followed by a 2 mL sterile preservative-free saline flush intracranially and clamping the EVD at the Becker to continuously transduce ICP. The EVD clamp trial period after drug injection was defined to be at least 30 min, but ideally is 60 min. This allows intracranial circulation of the drug using natural CSF pulsations within the intracranial compartment.

### 2.3. Study Rationale

After our initial 1 mg:1 mL IVN concentration study, our center adopted the use of IVN into clinical practice as a second line (rescue) agent to treat patients with symptomatic vasospasm after aSAH who were already receiving oral nimodipine per standard of care. In comatose patients with limited clinical examination, there had to be evidence of vasospasm on transcranial Doppler (TCD), a computed tomography angiogram (CTA), and CT perfusion to be deemed at high enough risk by the treating neurointensivist and neurosurgeon. Symptomatic vasospasm was defined as measurable clinical neurologic deficit with neuroimaging correlate of vasospasm on TCD, CTA, or cerebral angiogram or magnetic resonance imaging diffusion weighted image changes concerning for DCI and unrelated to initial SAH injury and any initial procedural complications. 

The rationale for developing the concentrated IVN form was based on our prior clinical experience that there are major volume injection limitations in a subset of patients with aSAH. These patients have small ventricular size as seen on CT or evidence of poor intracranial compliance/elastance by bedside ICP testing. Because many of these patients have a suboptimal amount of CSF that can be aspirated safely using the larger 1 mg:1 mL IVN dosing, this would prevent injection of a total 6 mL volume intracranially (4 mg:4 mL of nicardipine and 2 mL of sterile flush). In patients with poor intracranial compliance, even 1 to 2 mL injected intracranially can cause a major inflection point increase in ICP [17]. When the aspirated CSF volume is less than the injected volume, it can lead to dangerous spikes in ICP (e.g., >30 mm Hg) and comprised CPP (<55 mm Hg) [18]. Therefore, this non-isovolemic or net volume positive intracranial injection can lead to elevated ICP before 30 to 60 min and lead to unclamping of the EVD. Unclamping the EVD allows drug and CSF to efflux but also lessens ICP. Any drug coming back out into the EVD system is less drug effect time in the intracranial compartment. 

### 2.4. EVD Fenestration Ratio on CT as a Model for Drug Delivery

All patients with aSAH had an EVD inserted by neurosurgeons for symptomatic hydrocephalus, and the EVD tip position of the catheter was verified on subsequent noncontrast CT. An EVD was deemed adequate for intraventricular injection by the treating neurointensivist or vascular neurosurgeon when it was reviewed and showed at least half of the fenestrations inside the ventricle, prior to receiving any IVN injections. The method of visualizing CT to EVD fenestration ratio (ventricle/parenchyma) has been described previously [19]. The EVD fenestration ratio helps optimize intraventricular drug delivery (Figure 2) by ensuring the fenestrations are mostly within the CSF/ventricular compartment and can be a guide to minimize EVD tract hemorrhage during intraventricular drug injections [19,20]. 

### 2.5. IVN Concentrated Sterile Drug Preparation and IVN Injection Provider

All patients received central pharmacy–prepared sterile concentrated nicardipine 2.5 mg/mL mixed in sterile, preservative-free saline IVN. The central pharmacy delivered the drug to the intensive care unit (ICU) nursing bedside. Only neurocritical care or neurosurgery physicians and advanced practice providers (i.e., Advanced Practice Registered Nurse, Physician Assistant) signed off and supervised in sterile intraventricular injection procedures were permitted to perform the IVN procedure.

### 2.6. Safety and Tolerability of Adverse Events and CSF Utility Definitions

All patients underwent hourly neurologic assessments by neurologic ICU nursing, daily TCDs and, when symptomatic vasospasm suspected CTA with perfusion, a cerebral angiogram performed as needed per the treating vascular neurosurgeon. We defined *safety* using the following variables: central nervous system infection (e.g., bacterial meningitis and/or ventriculitis proven by microbiological culture) from repeated EVD injections, EVD tract hemorrhage after injections on subsequent brain imaging [19,20], and other clinical events related to intracranial injections such as headache if reported or neurologic examination decline was documented in the medical record. ICP elevations greater than 20 mm Hg after EVD injection were recorded, monitored, and managed by standard neurologic intensive care protocols and by reopening the EVD as defined above 22 mm Hg sustained for more than 2 min or a spike in ICP greater than 30 mm Hg in less than 1 min. We defined *tolerability* as any potential adverse drug reaction, such as systemic hypotension after injecting IVN, medication-induced rash, cardiovascular or vital sign changes on electrocardiogram monitoring, or IVN injection intolerance that requires permanent cessation of the injections per the neurointensivist attending’s clinical judgment. CSF *utility* was defined as the ability to draw CSF from an EVD during isovolemic injection procedures at least once a day. In younger patients or those with intraventricular hemorrhage or global brain edema with small ventricular size, drawing 6 mL of CSF may be impossible without creating high syringe suction pressure, which can potentially cause an EVD tract hemorrhage during frequent intrathecal/intraventricular drug procedures [19,20]. EVD tract hemorrhage can occur during EVD placement and is easily visible on noncontrast head CT and easily graded after EVD catheter injection procedures [19]. Electronic medical record data were recorded using Epic software (Epic Systems Corporation, Madison, WI, USA), which pulls data from the ICU vital signs and ICP recordings. All nurses validate ICP recordings, and they were trained on ICP zeroing and manual tidal techniques for ICP/CPP accuracy.

## 3. Results

### 3.1. Case 1

A 33-year-old man with no past medical history was transferred from another facility with aSAH (modified Fisher scale (mFS) grade 4, Hunt-Hess scale (HH) score 5). On examination upon arrival, his Glasgow Coma Scale (GCS) score was 4 (E1, V1T, M2), and head CT revealed extensive intraventricular cerebellar hemorrhage and aSAH with acute hydrocephalus and herniation (Figure 3). Head CTA showed right posterior fossa arteriovenous malformation. He underwent Onyx (Medtronic) embolization of the arteriovenous malformation followed by emergent decompressive suboccipital craniectomy. He had an EVD placed for symptomatic hydrocephalus, which was leveled and zeroed at the tragus and open to drain at 10 mm Hg or higher ICP pressures.

At admission, given the aSAH blood on CT, the patient was started on 60 mg nimodipine enterally via nasogastric tube every 4 h. Due to relative hypotension (systolic blood pressure <90 mm Hg), the standard nimodipine dosing schedule was reduced to 30 mg enterally every 2 h. However, he developed severe vasospasm in the bilateral middle cerebral artery ((MCA)-M1 segment mean flow velocity >200 cm/s) on post-aSAH day 7 as measured by TCD. Permissive hypertension (systolic blood pressure ≤180 mm Hg) was induced, as well as hydration for hemodynamic augmentation. CTA confirmed diffuse bilateral MCA-M1 intracranial vasospasm without evidence of infarction. Due to the patient’s poor neurologic examination (GCS 5, E2, V1T, M2), concentrated IVN (2.5 mg/mL) was injected every 8 h, as shown in Table 1, followed by a 2 mL sterile preservative-free 0.9% saline flush. If resistance was encountered during the injection of 2 mL, smaller flush volumes were administered. After each injection, the drainage system was turned off. ICP was monitored continuously with orders to reopen the EVD to drain CSF when ICP was sustained at 20 mm Hg or more for 2 min or more or if a large spike in ICP occurred (e.g., >30 mm Hg). ICP was measured continuously for 1 h after delivering the IVN and saline unless the ICP drain was opened. Twenty-six concentrated IVN injections were administered to this patient without adverse events (Table 1).

The hospital course was prolonged with initial aspiration pneumonia requiring a percutaneous tracheostomy, an endoscopic third ventriculostomy, a ventriculoperitoneal shunt, and a percutaneous endoscopic gastrostomy tube. He was discharged on post-aSAH day 42 to a rehabilitation facility with a modified Rankin Scale (mRS) score of 4. At 3-month follow-up, his mRS was 3.

### 3.2. Case 2

A 36-year-old woman with a past medical history of sickle cell anemia, chronic back pain, and headaches presented with aSAH (mFS grade 4, HH score 4) (Figure 4). On arrival, her GCS score was 3 (E1, V1, M1), and she had an unreactive 7 mm right pupil. Head CT and CTA revealed diffuse aSAH blood, blood in the fourth ventricle, and obstructive hydrocephalus with a basilar tip aneurysm. The epicenter of the aSAH was predominately around the midbrain, suggesting the basilar aneurysm as the source of bleeding (Figure 2). The patient was intubated, and she underwent cerebral angiography with stent-assisted coiling of the basilar tip aneurysm. Bilateral EVDs were placed, and she was started on 60 mg of nimodipine every 4 h enterally via nasogastric tube. However, due to hypotension, the patient’s nimodipine dose was reduced to 30 mg every 2 h. On post-aSAH day 5, she was noted with vasospasm on TCDs (left proximal MCA 24.2 cm/s and right MCA 24.8 cm/s). CTA revealed M1 stenosis, and CT perfusion showed watershed distribution infarcts. The patient was started on concentrated IVN (2.5 mg/mL) every 8 h, with total volumes and pre-treatment and post-treatment ICPs shown in Table 2. Thirteen concentrated IVN injections were completed without any adverse effects.

The patient’s hospital course was complicated by sepsis, percutaneous endoscopic gastrostomy tube placement, and tracheostomy. On post-aSAH day 30, she was discharged to a long-term acute care facility with an mRS score of 5. However, we later learned in reviewing the medical record in this case series that the patient died after discharge to the long-term acute care facility.

### 3.3. Case 3

A 70-year-old woman with a past medical history of hypertension, hyperlipidemia, and diabetes mellitus type 2 presented to the emergency department with aSAH (mFS grade 3, HH 4) (Figure 5). On arrival, her systolic blood pressure was above 190 mm Hg, and her National Institutes of Health Stroke Scale score was 9. Head CT showed left frontal intraparenchymal hemorrhage and SAH predominantly in the left sylvian fissure with extension over the left cerebral convexity. The bleed was secondary to the left internal carotid artery terminus aneurysm (1.5 × 1.5 mm), with dysplastic aneurysmal dilation of the distal left M1 extending in both left M2 segments. The patient underwent left craniotomy for clipping of the ruptured aneurysm as well as wrapping of the left MCA dysplastic aneurysm. An EVD was placed, which opened at 20 mm Hg of ICP. The patient received 60 mg of nimodipine every 4 h through a nasogastric feeding tube. Her blood pressure was labile throughout the admission, at times requiring medical treatment for hypertension and at other times intravenous vasopressors for hypotension. The patient received concentrated IVN (2.5 mg/mL) for vasospasm, receiving a total of 20 concentrated injections without any adverse effects (Table 3). She had a prolonged hospital course and was discharged to a long-term rehabilitation facility with a GCS score of 11, National Institutes of Health Stroke Scale score of 19, and mRS score of 3.

## 4. Discussion

Despite considerable research in SAH therapeutics, there remains only nimodipine in randomized, placebo-controlled trials for improving neurologic outcomes. In patients with aSAH with an existing EVD, IVN is an attractive option, especially in those with dose-reduced nimodipine who do not tolerate the standard dose of nimodipine 60 mg every 4 h and develop symptomatic vasospasm or CTA and CT perfusion evidence of severe vasospasm-related ischemia. An individualized medicine approach to prevent aSAH secondary neurologic injury is desperately needed, one that can enhance standard of care nimodipine use. While emergent cerebral angiography, angioplasty of vasospastic arteries, and intra-arterial common carotid artery injection are often used in patients with aSAH with symptomatic vasospasm, cerebral angiograms are invasive and more costly than injecting 4 mg of nicardipine into an existing EVD. However, not all patients with aSAH have an EVD, which limits this approach, and intraventricular injections are not without some inherent risks of infection and bleeding [19,20,21]. When an EVD is placed per standard of care in aSAH, however, it can become a vehicle for diagnostic discovery and therapeutic intervention in this population. An EVD also allows for CSF sampling and ICP control via CSF diversion and drainage, ICP monitoring [22], and cerebral perfusion measurement and could serve as a route for future intrathecal drug delivery.

Previous studies have investigated the use of intrathecal nicardipine for DCI prophylaxis [11,23,24]. In a study by Shibuya et al. [24] 50 patients with aSAH, all with mFS grade 3 and aneurysm clipping, received 2 mg of nicardipine dissolved in 10 mL of saline administered every 8 h for 10 days. A decrease in incidence of angiographic (20%) and symptomatic vasospasm (26%) was observed in patients receiving intrathecal nicardipine and increased good clinical outcomes by 15% at 1 month. Two patients suffered from meningitis but were successfully treated [24]. Goodson et al. [11] described a regimen of 4 mg of intrathecal nicardipine dissolved in 10 mL of saline every 12 h during postoperative days 3 to 14 for 177 patients with mFS grade 3 following aSAH. Twenty patients (11.3%) showed angiographic vasospasm, and 10 (5.7%) of these showed symptomatic vasospasm (6 transient and 4 permanent). At 6 months, 89.2% of patients had positive outcomes (assessed as good recovery and moderately disabled using Glasgow outcome scale) [11]. Similarly, in a study by Fujiwara et al. [23] five patients with aSAH and mFS grades of at least 3 were given continuous nicardipine through an infusion pump connected to an intracranial cisternal drain, with 12 mL administered daily (8 mg of nicardipine) for 14 days. All five cases had good recovery at 3 months. The authors suggested that this method of continuous drug infusion through cisternal drain has the advantage of ease over other conventional methods, although it involves an increased risk of infection [23].

Other studies have demonstrated the safety and efficacy of IVN for the treatment of vasospasm [12,25], although with varying doses (mg), frequency of dosing, and reported outcomes. In a case-control study, 14 cases treated with IVN with aSAH (including one with arteriovenous malformation) with mFS grades of 3 to 4 were matched with controls and given a median (range) dose of 4 (3–7) mg IVN (1 mg:1 mL concentration) with a median (range) of 7 (1–17) doses [12]. They reported an improvement in cerebral blood flow on TCD but failed to show mRS and Glasgow outcome scale score improvement over controls by 90 days. IVN was well tolerated with no adverse events [12]. Although the results of these studies suggest that intrathecal (intraventricular) administration of nicardipine can treat immediate vasospasm on neuroimaging, they all lack randomized, placebo-controlled trials on aSAH outcomes. These retrospective cohort and case-controlled studies are also limited due to the small number of patients, variable protocols of IVN administration, and lack of randomization.

To the best of our knowledge, this is the first report on the use of concentrated IVN (2.5 mg/mL) nicardipine using only 4 mg concentrated into a 1.6 mL volume of injected drug in a subset of poor grade SAH with no adverse events or development of central nervous system infection. We report the ICP pre- and postprocedure measurements using this concentrated dose, which are not reported in prior IVN studies. We did not observe any obvious drug reactions, immediate hypotension, systemic rashes, or other signs of intolerance. Despite the small size of this sample, we feel that the total number of concentrated IVN injections provides some value, especially in those patients who cannot tolerate a 1 mg:1 mL formulation, which is 6 to 8 mL CSF aspiration and similar volume injection. Compared to other reported IVN injection methods, the main advantage of this is the high concentration of nicardipine at 2.5 mg/mL, which allows 4 mg to be given in only 1.6 mL and with an intracranial flush of 2 mL, totaling only 3.6 mL for the entire isovolemic technique.

Further, aSAH cases with borderline intracranial compliance (dV/dP) or high elastance (dP/dV) [22] or small intraventricular volumes for whatever cause can benefit from this smaller isovolemic intracranial CSF injection methodology. Intraventricular injection of small volumes are also typically better tolerated than larger volumes, since even small volumes injected intracranially can drastically increase ICP [15]. Therefore, concentrated IVN could be particularly helpful in young patients with relatively small ventricular volumes compared to older patients with focal or global brain edema pathophysiology causing secondary or relatively small ventricles. Similarly, some aSAH cases have so-called slit ventricles (i.e., small/tiny lateral ventricles), in which large isovolemic aspiration and intraventricular injections can adversely raise ICP and reduce CPP [17].

We acknowledge this study has several limitations. First, the overall sample size is a small (N = 59) number of total injections in a small number of patients with poor-grade aSAH. The data presented only represent preliminary safety and tolerability data for a concentrated IVN drug dosing. However, we feel overall the data are important to report given the cerebrovascular physiologic limitations of injecting higher volumes intracranially in this patient population, which can and often does compromise CPP or lead to unclamping of the EVD after drug injection. We did not randomize patients to a placebo arm since our practice has adopted the use of IVN with success in patients with aSAH developing symptomatic vasospasm or DCI and by neuroimaging defined targets (TCD and CTA). We admit this introduces a potential bias. However, we feel this is one that warrants a future larger randomized trial that can be built upon this safety and tolerability data. In addition, despite having only a small sample size of patients, each received 13 to 26 IVN injections. We also acknowledge our retrospective design introduces many other limitations in design, and we cannot provide causation, only make limited observations or associations. Despite all these limitations, the data were recorded utilizing an electronic medical record with a well-documented system and included well-documented structured procedural notes about aspiration volume, injection volume, and any other issues. Further, the electronic medical record included well-documented clinical notes and hourly ICU vital signs. Overall, we acknowledge these limitations, which can only be addressed in a future Phase I/II multicenter, randomized, placebo-controlled prospective trial with a focus on patient outcomes. Such a trial will require considerable standardization of second-line and third-line symptomatic vasospasm and DCI interventions as well as neuromonitoring and neuroimaging defined endpoints.

## 5. Conclusions

A small dose (1.6 mL) of concentrated IVN (4 mg total at 2.5 mg/mL) can be used for aSAH-related vasospasm and appears relatively safe and tolerable. This is especially true in patients with poor-grade SAH, in whom the intraventricular volume is relatively small, and compromise of ICP or CPP may develop with larger volumes. A multicenter, randomized, placebo (saline)-controlled trial is warranted to validate these findings and test the hypothesis that concentrated IVN could potentially improve neurologic and neuroimaging DCI outcomes in patients with aSAH.

## Figures and Tables

**Figure 1 jpm-13-00428-f001:**
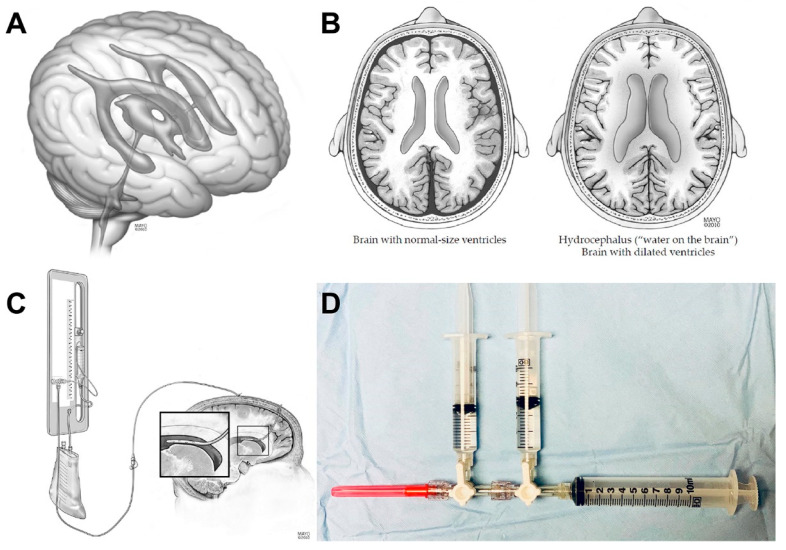
(**A**,**B**) Ventricular anatomy in aneurysmal subarachnoid hemorrhage. Hydrocephalus can occur from obstructive or nonobstructive (communicating) patterns, which benefit from external ventricular drain (EVD) placement, cerebral spinal fluid (CSF) diversion, and intracranial pressure monitoring. (**C**) EVD Becker drainage system, location of EVD catheter in the lateral ventricle, and location of typical intraventricular nicardipine injection. (**D**) Three-way stopcock EVD injection apparatus for sterile isovolemic drug injection. Red 18G needle (bottom left) for insertion into the EVD rubber port to inject/aspirate. Left syringe: Intraventricular nicardipine (drug) injection port. Middle syringe: Sterile preservative-free saline flush. Right syringe: Large 10 mL syringe used to aspirate CSF for isovolemic procedure, “Volume out must equal volume in”. Used with permission of Mayo Foundation for Medical Education and Research. All rights reserved.

**Figure 2 jpm-13-00428-f002:**
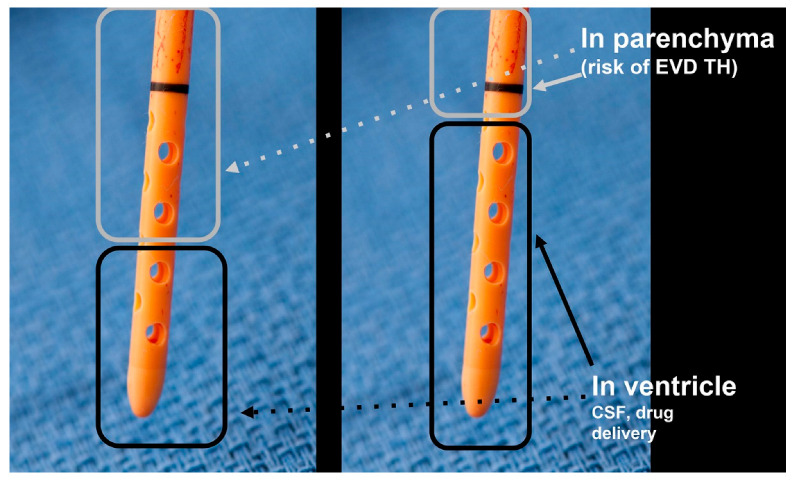
An explanted external ventricular drain (EVD) visually demonstrates the rationale for intraventricular nicardipine delivery and how the EVD (ventricle/parenchyma) fenestration ratio can help guide intrathecal drug injection safety. EVD fenestrations can be seen on computed tomography bone windows when using large bore 1.9 mm inner diameter EVD catheters, such as those used in this study.

**Figure 3 jpm-13-00428-f003:**
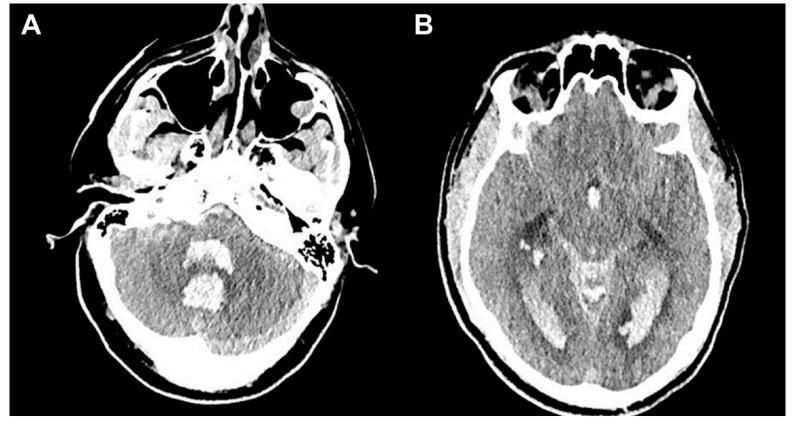
Case 1: Head Computed Tomography. (**A**) Cerebellar hemorrhage and adjacent markedly dilated fourth ventricle with intraventricular hemorrhage (IVH). (**B**) Bilateral lateral and posterior ventricular system IVH with dilated temporal and lateral ventricles consistent with acute hydrocephalus.

**Figure 4 jpm-13-00428-f004:**
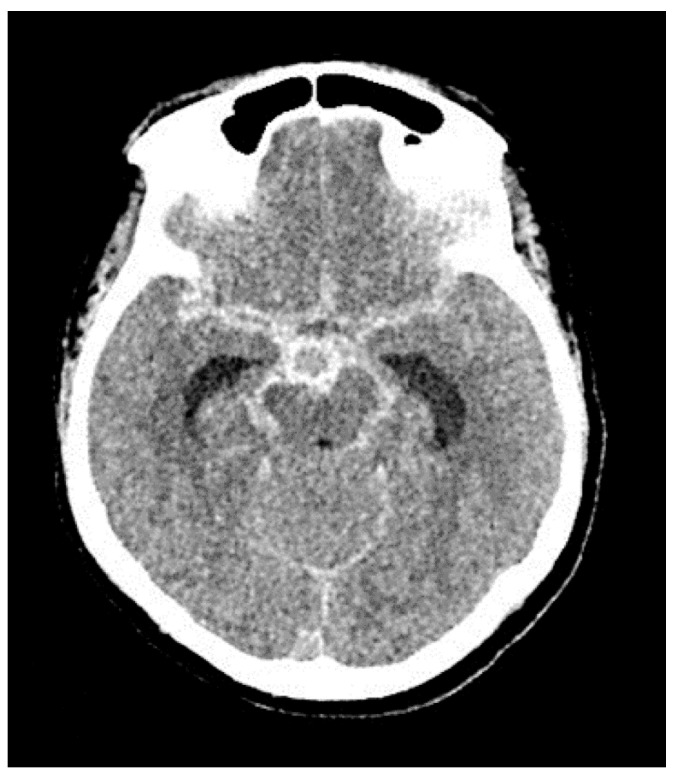
Case 2: Head Computed Tomography Showing Subarachnoid Hemorrhage Predominantly Around the Midbrain.

**Figure 5 jpm-13-00428-f005:**
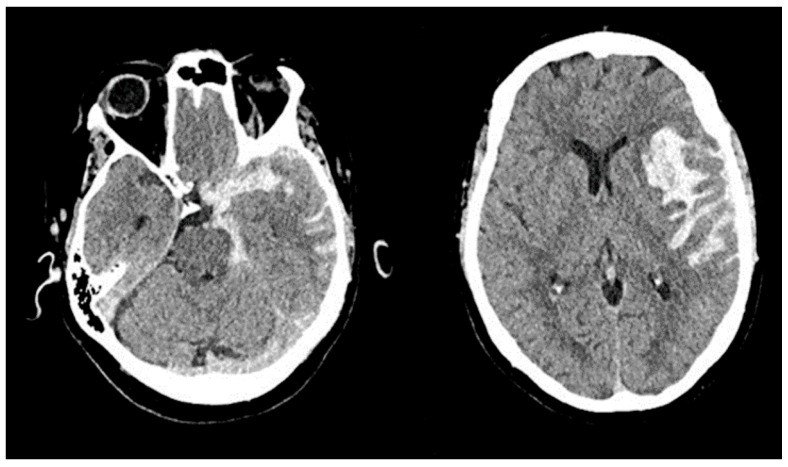
Case 3: Head Computed Tomography Showing Frontal Intraparenchymal Hemorrhage with Left Sylvian Fissure Extension.

**Table 1 jpm-13-00428-t001:** Case 1: IVN Dosage and Related ICP Readings.

Dose Number	CSF Volume Aspirated, mL	Volume 2.5 mg/mL IVN Solution, mL	Volume Saline Flush, mL	Isovolemic Goal, mL ^a^	Pre-Treatment ICP, mm Hg	Post-Treatment ICP, mm Hg
1	4.5	2.5	2.0	Yes, 0.0	4	2
2	4.5	2.5	2.0	Yes, 0.0	9	8
3	2.4	1.6	0.8	Yes, 0.0	5	6
4	5.0	1.6	1.4	No, −2.0	8	7
5	5.0	1.6	2.0	No, −1.4	10	12
6	3.5	1.6	2.0	No, +0.1	8	7
7	1.2	1.2	1.0	No, +1.0	11	11
8	3.0	1.6	1.5	No, +0.1	9	15
9	4.0	1.6	2.4	Yes, 0.0	10	11
10	4.0	1.6	2.4	Yes, 0.0	14	11
11	4.0	1.6	1.4	No, −1.0	5	4
12	3.0	1.6	1.4	Yes, 0.0	9	7
13	3.0	1.6	1.4	Yes, 0.0	8	13
14	2.0	2.0	1.0	No, +1.0	8	9
15	3.0	1.6	1.4	Yes, 0.0	8	10
16	2.8	1.6	1.4	No, +0.2	6	7
17	3.0	1.6	1.5	No, +0.1	7	5
18	3.0	1.6	1.5	No, +0.1	4	4
19	3.0	1.6	1.4	Yes, 0.0	8	6
20	3.0	1.6	1.4	Yes, 0.0	8	6
21	4.0	1.6	2.4	Yes, 0.0	7	9
22	6.0	1.6	4.4	Yes, 0.0	15	14
23	6.0	1.6	2.0	No, −2.4	12	12
24	4.0	1.6	2.4	Yes, 0.0	9	10
25	4.0	1.6	2.4	Yes, 0.0	6	10
26	4.0	1.6	2.4	Yes, 0.0	1	1

Abbreviations: CSF, cerebrospinal fluid; ICP, intracranial pressure; IVN, intraventricular nicardipine. ^a^ Isovolemic goal = injected volume total − aspirated CSF volume.

**Table 2 jpm-13-00428-t002:** Case 2: IVN Dosage and Related ICP Readings.

Dose Number	CSF Volume Aspirated, mL	Volume 2.5 mg/mL IVN Solution, mL	Volume Saline Flush, mL	Isovolemic Goal, mL ^a^	Pre-Treatment ICP, mm Hg	Post-Treatment ICP, mm Hg
1	4.0	1.6	2.0	No, −0.4	8	11
2	2.0	1.6	1.0	No, +0.6	15	18
3	2.5	1.6	2.0	No, +1.1	7	10
4	6.0	1.6	2.0	No, −2.4	NA	5
5	5.0	1.5	2.0	No, −1.5	6	8
6	5.0	1.6	3.4	Yes, 0.0	15	16
7	4.0	1.6	2.0	No, −0.4	10	6
8	4.0	1.6	2.0	No, −0.4	16	16
9	4.0	1.6	2.0	No, −0.4	12	10
10	2.0	1.6	0.4	Yes, 0.0	12	10
11	2.0	1.6	0.4	Yes, 0.0	11	11
12	4.0	1.6	2.4	Yes, 0.0	6	6
13	4.0	1.6	2.4	Yes, 0.0	11	10

Abbreviations: CSF, cerebrospinal fluid; ICP, intracranial pressure; IVN, intraventricular nicardipine; NA, not available. ^a^ Isovolemic goal = injected volume total − aspirated CSF volume.

**Table 3 jpm-13-00428-t003:** Case 3: IVN Dosage and Related ICP Readings.

Dose Number	CSF Volume Aspirated, mL	Volume 2.5 mg/mL IVN Solution, mL	Volume Saline Flush, mL	Isovolemic Goal, mL ^a^	Pre-Treatment ICP, mm Hg	Post-Treatment ICP, mm Hg
1	4.0	1.6	2.0	No, −0.4	7	7
2	4.0	1.6	2.0	No, −0.4	10	7
3	4.0	1.6	2.0	No, −0.4	10	11
4	3.6	1.6	2.0	Yes, 0.0	7	5
5	5.0	1.6	1.0	No, −2.4	11	10
6	6.0	1.6	2.4	No. −2.0	3	5
7	4.0	1.6	2.4	Yes, 0.0	12	14
8	4.0	1.6	2.4	Yes, 0.0	8	9
9	5.5	1.6	2.0	No, −1.9	7	6
10	4.0	1.6	2.5	No, +0.1	7	8
11	5.0	1.6	3.4	Yes, 0.0	7	5
12	5.5	1.6	2.0	No, −1.9	2	6
13	4.0	1.6	2.5	No, +0.1	6	8
14	4.0	1.6	2.5	No, +0.1	2	8
15	5.0	1.6	3.4	Yes, 0.0	4	3
16	3.6	1.6	2.0	Yes, 0.0	9	9
17	3.6	1.6	2.0	Yes, 0.0	9	10
18	3.6	1.6	2.0	Yes, 0.0	12	12
19	4.0	1.6	2.4	Yes, 0.0	2	1
20	NA	1.6	2.0	NA	13	7

Abbreviations: CSF, cerebrospinal fluid; ICP, intracranial pressure; IVN, intraventricular nicardipine; NA, not available. ^a^ Isovolemic goal = injected volume total − aspirated CSF volume.

## Data Availability

Data is unavailable due to privacy or ethical restrictions.

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
