# Peer review of "Safety and Tolerability of Concentrated Intraventricular Nicardipine for Poor-Grade Aneurysmal Subarachnoid Hemorrhage–Related Vasospasm"

_jpm, 2023, doi:10.3390/jpm13030428_

Round 1

Reviewer 1 Report

Dear Authors,

I am glad to have the opportunity to review your work. This study aimed to report the preliminary safety, tolerability, and cerebral spinal fluid sampling utility institutional of serial injections of concentrated intraventricular nicardipine in the treatment of aneurysmal subarachnoid hemorrhage.

The study is poorly designed, materials and methods are not written well.

My concerns and questions for the authors are following:

1.      There is no proper Methodology section

2.      Please add what type of study this is, when it took place, inclusion, exclusion criteria, variables…

3.      In the results section, the figures only present imaging showing ASAH. It needs to be written, on which day (since the ASAH) the imaging was done. Also, you need to add imaging after the treatment with intraventricular nicardipine, and also follow-up (discharge or later on)

4.      The number of cases is too low, please add at least two more

5.      Based on what you decided that IVN treatment is relatively safe, when one out of three patients you presented, died

Therefore, I recommend mayor revision of the paper.

Author Response

Dear Authors,

I am glad to have the opportunity to review your work. This study aimed to report the preliminary safety, tolerability, and cerebral spinal fluid sampling utility institutional of serial injections of concentrated intraventricular nicardipine in the treatment of aneurysmal subarachnoid hemorrhage.

The study is poorly designed, materials and methods are not written well. 

My concerns and questions for the authors are following:

  1. There is no proper Methodology section
  • Thank you for your comment. The section titled “IV Concentration and Injection Methodology” describes the methodology.
  1. Please add what type of study this is, when it took place, inclusion, exclusion criteria, variables…
  • Thank you for this recommendation, we added in the inclusion criteria of what patients we included in this retrospective chart review limited risk study.
  1. In the results section, the figures only present imaging showing ASAH. It needs to be written, on which day (since the ASAH) the imaging was done. Also, you need to add imaging after the treatment with intraventricular nicardipine, and also follow-up (discharge or later on)
  • Thank you for this suggestion of including the day of that imaging was performed. Unfortunately, this information is not available to include in this manuscript but is absolutely something we would keep in mind with future publications.
  1. The number of cases is too low, please add at least two more
  • Thank you for this suggestion. At this time, these are the only cases available that concentrated Nicardipine was used. We agree, having more cases would add to the manuscript.
  1. Based on what you decided that IVN treatment is relatively safe, when one out of three patients you presented, died
  • Thank you for this comment. The patients in the manuscript did not have complications from the IT Nicardipine, however with the natural disease process of SAH with a high morbidity rate, one patient did pass away after hospital discharge.

Therefore, I recommend mayor revision of the paper.

Thank you, we have made adjustments based on your recommendations where we were able too.

Reviewer 2 Report

This is a well-structured article that needs to be published. The rationale behind the article prompts worldwide attention for more innovative research on the subject matter.

No weakness was noted. Well-elaborated conclusion with cutting-edge ideas for future researchers. The conclusion is consistent with evidence-based medicine. All errors pointed out must be corrected before publication.

Author Response

REVIEWER 2

This is a well-structured article that needs to be published. The rationale behind the article prompts worldwide attention for more innovative research on the subject matter.

No weakness was noted. Well-elaborated conclusion with cutting-edge ideas for future researchers. The conclusion is consistent with evidence-based medicine. All errors pointed out must be corrected before publication.

  • Thank you for your comments and kind words. We appreciate you taking the time to review our work.

Reviewer 3 Report

The reviewed manuscript entitled ‘Safety and Tolerability of Concentrated Intraventricular Nicardipine for Poor-Grade Aneurysmal Subarachnoid Hemorrhage-Related Vasospasm’ written by Kaneez Zahra et al. provides interesting results regarding the alternative form of nicardipine administration to prevent vasospasm in three patients with poor-grade aneurysmal subarachnoid hemorrhage. The authors evaluated the safety and tolerability of the described technique and no major adverse events were observed. I have only minor comments for this manuscript:

11. Human subjects were qualified for the experiments, but there is no information about ethical aspects of the study (e.g., identificator of ethical commission consent). The inclusion and exclusion criteria for the qualification of patients could be more highlighted.

22. In the legend of Figure 2, it should be clarified which of the presented options is better and for what reason. The magnification of the pictures should be provided, if applicable.

33. In the description of case 3, the occurrence or not of adverse effects should be highlighted.

44. The tables present data about dosage, ICP parameters and the levels of isovolemic goal achievement. Did the authors try to find any relationships between these variables? Especially, do CSF volume, saline flush volume, and isovolemic goal parameters influence posttreatment ICP? It seems to be especially interesting because of the application of a more concentrate solution of nicardipine (lower volumes) in the described method.

55. Table gathering the clinical characteristics of patients (e.g., age, sex, mFS grade, HH score), during admission and when discharge, could be provided for a better view of the study group.

I believe that my suggestions will be helpful to the authors to increase the quality of the reviewed manuscript.

Author Response

The reviewed manuscript entitled ‘Safety and Tolerability of Concentrated Intraventricular Nicardipine for Poor-Grade Aneurysmal Subarachnoid Hemorrhage-Related Vasospasm’ written by Kaneez Zahra et al. provides interesting results regarding the alternative form of nicardipine administration to prevent vasospasm in three patients with poor-grade aneurysmal subarachnoid hemorrhage. The authors evaluated the safety and tolerability of the described technique and no major adverse events were observed. I have only minor comments for this manuscript:

  1. Human subjects were qualified for the experiments, but there is no information about ethical aspects of the study (e.g., identificator of ethical commission consent). The inclusion and exclusion criteria for the qualification of patients could be more highlighted.
  • Thank you for this suggestion. This was a retrospective review that was considered
  1. In the legend of Figure 2, it should be clarified which of the presented options is better and for what reason. The magnification of the pictures should be provided, if applicable.
  • Thank you for your recommendation on the magnification of this image. We are able to provide magnification of this picture if needed for publication.
  1. In the description of case 3, the occurrence or not of adverse effects should be highlighted.
  • Thank you for this recommendation, we have added in verbage to highlight that no adverse effects were noted after the 20 injections in Case # 3.
  1. The tables present data about dosage, ICP parameters and the levels of isovolemic goal achievement. Did the authors try to find any relationships between these variables? Especially, do CSF volume, saline flush volume, and isovolemic goal parameters influence posttreatment ICP? It seems to be especially interesting because of the application of a more concentrate solution of nicardipine (lower volumes) in the described method.
  2. Table gathering the clinical characteristics of patients (e.g., age, sex, mFS grade, HH score), during admission and when discharge, could be provided for a better view of the study group.
  • Thank you for this suggestion of making a table to highlight the clinica characteristics of patients, we have included the characteristics of the patients in the article.

I believe that my suggestions will be helpful to the authors to increase the quality of the reviewed manuscript.